# Exogenous Nucleotides Improved the Oxidative Stress and Sirt-1 Protein Level of Brown Adipose Tissue on Senescence-Accelerated Mouse Prone-8 (SAMP8) Mice

**DOI:** 10.3390/nu14142796

**Published:** 2022-07-07

**Authors:** Xiujuan Wang, Rui Liu, Chan Wei, Meihong Xu, Yong Li

**Affiliations:** Department of Nutrition and Food Hygiene, School of Public Health, Peking University, Beijing 100191, China; 1510306116@pku.edu.cn (X.W.); liuruipku@163.com (R.L.); chanwei2018@126.com (C.W.); xumeihong@bjmu.edu.cn (M.X.)

**Keywords:** nucleotides, brown adipose tissue, AMPK/Sirt-1, oxidative stress, aging

## Abstract

Brown adipose tissue (BAT) is of great importance in rodents for maintaining their core temperature via non-shivering thermogenesis in the mitochondria. BAT′s thermogenic function has been shown to decline with age. The activation of adenosine 5′-monophosphate (AMP)-activated protein kinase/sirtuin-1 (AMPK/Sirt-1) is effective in regulating mitochondrial function. Exogenous nucleotides (NTs) are regulatory factors in many biological processes. Nicotinamide mononucleotide (NMN), which is a derivative of NTs, is widely known as a Sirt-1 activator in liver and muscle, but the effect of NMN and NTs on aging BAT has not been studied before. The purpose of this study was to investigate the effect of NTs on aging senescence-accelerated mouse prone-8 (SAMP8) mice. Senescence-accelerated mouse resistant 1 (SAMR1) mice were set as the model control group and NMN was used as the positive control. Male, 3 month old SAMP8 mice were divided into the SAMP8-normal chow (SAMP8-NC), SAMP8-young-normal chow (SAMP8-young-NC), NMN, NTs-free, NTs-low, NTs-medium, and NTs-high groups for long-term feeding. After 9 months of intervention, interscapular BAT was collected for experiments. Compared to the SAMP8-NC, the body weight and BAT mass were significantly improved in the NT-treated aging SAMP8 mice. NT supplementation had effects on oxidative stress in BAT. The concentration of malondialdehyde (MDA) was reduced and that of superoxide dismutase (SOD) increased significantly. Meanwhile, the expression of the brown adipocyte markers uncoupling protein-1 (UCP-1), peroxisome proliferator-activated receptor-γ coactlvator-1α (PGC-1α), and PR domain zinc finger protein 16 (PRDM16) were upregulated. The upregulated proteins may be activated via the Sirt-1 pathway. Thus, NT supplementation may be helpful to improve the thermogenesis of BAT by reducing oxidative stress and activating the Sirt-1 pathway.

## 1. Introduction

With advances in public health and medical science, the proportion of people over 65 year old continues to rise. The increase in the aging population aggravates the burden of chronic disease. Adipose tissue is important for energy storage and endocrine functions and is associated with chronic diseases including cardiovascular disease and diabetes [1]. Improving the function of aging adipose tissue is of great significance for reducing age-related metabolic dysfunction and senile diseases [2]. Brown adipose tissue (BAT) is formed by the aggregation of multivesicular brown adipocytes. There are numerous small lipid droplets and abundant mitochondria scattered in stimulated brown adipocytes. Brown adipocytes uniquely express uncoupling protein-1 (UCP-1) in the mitochondrial intima, which produces heat in a process called non-shivering thermogenesis [3]. A previous study has shown that activated BAT plays an important role in systemic energy consumption, human glucose balance, and insulin sensitivity by consuming glucose and free fatty acids (FFAs) for thermogenesis [4]. With an increase in age, the accumulation of adverse factors promotes BAT aging accompanied by a decrease in total mass, mitochondrial function, and UCP-1 activation [5]. Macrophage infiltration and excessive triglyceride accumulation promote an increase in the white adipocyte-like phenotype in BAT and these gradually replace brown adipocytes; that is, the brown adipocytes “whitening” [6,7]. Maintaining the brown-like adipocyte morphology and function of BAT is conducive to preventing metabolic diseases related to age and obesity [8].

The reactive oxygen species (ROS) produced by mitochondria acts as an important signal molecule in BAT under normal physiological levels. At biological levels, ROS induce the production of UCP-1 by enhancing the cyclic adenosine monophosphate (cAMP)/p38 mitogen-activated protein kinase signal. Maintaining a certain amount of ROS is conducive to maintaining the normal physiological metabolism and function of BAT [9]. As a central mediator of the cellular response to energetic stress, AMPK regulates cell growth and other important cellular processes including lipid and glucose metabolism [10]. Sirt-1 and AMPK have a close interaction in the regulation of energy, metabolism, and aging, since they can reciprocally enhance each other′s activity. AMPK enhances Sirt-1 activity by increasing cellular nicotinamide adenine dinucleotide (NAD) levels, resulting in deacetylation of PGC-1α, the downstream target of Sirt-1. Sirt-1 activates liver kinase B1 (LKB1), leading to an increase in pAMPK [11]. The AMPK/Sirt-1 pathway assists in maintaining cell mitochondrial function, and it shows broad application prospects in the areas of anti-inflammation, antioxidants, maintaining mitochondrial function, and regulating apoptosis.

Nucleotides (NTs) are subunits of nucleic acids that take part in many complex physiological processes including energy metabolism, intercellular signal transduction, enzymatic reactions, and regulating the cell cycle as a synthetic nitrogen source. Exogenous NTs are treated as conditionally essential nutrients. When the body suffers from intestinal injury, rapid growth, or reduced protein intake, it is necessary to obtain additional NTs from the diet to meet these special needs. Exogenous NTs have various biological functions such as immune regulation [12], promoting the growth and development of infants [13], protecting gastrointestinal [14] and liver function [15], improving memory [16], antioxidation [17], and regulating lipid metabolism [18]. Nucleotide derivatives, such as nicotinamide mononucleotide (NMN), have been proven to improve mitochondrial function by activating Sirt-1 and playing an important role in delaying aging [19]. However, there is no study showing that NTs can be effective against BAT aging.

In this study, we focused on the effect of NTs on BAT in aging senescence-accelerated mouse prone-8 (SAMP8) mice to explore whether NTs are helpful to reduce the stage of oxidative stress and improve the expression of brown adipocyte markers. The senescence-accelerated mouse (SAM) represents a group of inbred mouse strains developed through the backcrossing of AKR/J mice and subsequent phenotyping as a model for the study of human aging and age-related diseases. The SAM strain consists of nine senescence-prone (SAMP) strains and three senescence-resistant (SAMR) strains [20]. Routine postmortem examinations and a series of systematically designed studies have revealed that SAMP strain mice exhibit “accelerated senescence” and “normal aging” [21]. Traditional rodent animal models take at least 18 months to develop aging phenotypes, but SAMP8 shortens the process of aging and develops adipocyte hypertrophy and ectopic lipid accumulation in liver and muscle at 40 weeks old, while SAMR1 and C57BL/6 do not have metabolic disorders at this age [22]. We raised the mice from 3 months old for 9 months and took 12 month old mice as the intervention endpoint; thus, we could observe the effect of NTs on aging BAT.

## 2. Materials and Methods

### 2.1. Materials

Exogenous NTs (5′AMP:5′CMP:5′GMPNa_2_:5′UMPNa_2_ = 16:41:19:24) and NMN compound were extracted from cane sugar by enzymatic hydrolysis, provided by Zhen-Ao Biotechnology Ltd., Co., (Dalian, China). To simulate the way that people intake NTs in general, we customized the food with different doses of NTs from Beijing Keao Xieli Feed Co., Ltd., (Beijing, China). The levels of NTs in purified and standard food can be seen in Table 1.

### 2.2. Animals and Treatment

Male SAMP8 and SAMR1 (3 month old) mice were provided by the laboratory animal research center, Peking University. After 1 week of adaptation, SAMP8 mice were randomly divided into 6 groups: SAMP8-NC (normal chow), NTs-free (NTs-F), NTs-low (NTs-L), NTs-medium (NTs-M), NTs-high (NTs-H), and NMN. SAMR1 mice were set-up as the model group (SAMR1-NC). Three-month-old SAMP8 mice in the SAMP8-young-NC group were included when the other mice were at the age of 9 months. The mice in the SAMP8-NC, SAMR1, and SAMP8-young-NC groups were fed with standard food (American Institute of Nutrition Rodent Diets-93M (AIN-93M diet)). The mice in the NTs-F group were fed with the purified formula of the AIN-93M diet. The NTs-L, NTs-M, and NTs-H groups were fed by adding different doses of exogenous NTs to the standard food. The doses of the intervention groups and the schematic diagram of the animal experiment are shown in Table 2. The SAMP8-young-NC mice SAMP8-young-NC were 6 months old at the intervention endpoint, and all other groups were 12 months old at the intervention endpoint. Every mouse was caged alone at 24 ± 2 °C, 50–60% relative humidity, and with a 12 h/12 h light/dark cycle. All mice were given free access to water and food. The food intake and weight of the mice were measured and recorded each week. After the long-term feeding interventions, body composition was detected and interscapular BAT was carefully and immediately collected and stored at −80 °C for the experiments.

### 2.3. Body Composition Detection

At the intervention endpoint, body composition was assessed using nuclear magnetic resonance (EchoMRI-700, Houston, TX, USA). The total fat mass (g) of mice was obtained by automatic scanning. We repeated the measurement three times and discarded measurement data with large errors. The average value of the measurement was taken as the total body fat mass of the mice.

### 2.4. Detection of Oxidative Stress-Associated Biological Indicators

A 10% tissue homogenate was prepared for test kit detection. The test kits used for oxidative stress involved a malondialdehyde (MDA) assay kit (thiobarbituric acid (TBA) method), a total superoxide dismutase (SOD) assay kit with WST-8, a catalase (CAT) assay kit (visible light), and a glutathione peroxidase (GSH-Px) assay kit (colorimetric method), which were all purchased from the Nanjing Jiancheng Bioengineering Institute. The biological indicators were measured according to the protocols provided with the assay kits.

### 2.5. Western Blot Analysis

Three mice in each group were randomly selected for Western blot analysis. Total proteins were extracted and measured using a BCA Protein Assay Kit. Gel electrophoresis was used to separate the lysates, which were then transferred to PVDF membranes. A total of 5 g of skimmed milk powder was dissolved in 100 mL TBST using as a blocking agent for 4 h at room temperature. After blocking, the membranes were incubated with primary antibodies overnight at 4 °C; UCP-1 (E9Z2W) XP rabbit mAb, p-AMPKα, AMPKα rabbit mAb, and sirt-1 rabbit mAb were obtained from CST. Anti-PGC1 alpha, anti-PPAR alpha, and anti-vasfatin were purchased from Abcam. Human/mouse PR domain zinc finger protein 16 (PRDM16) antibody was purchased from R&D. Then, the membranes were incubated with secondary antibodies for 4 h at 4 °C. We visualized the protein band using enhanced chemiluminescence (ECL) detection kits.

### 2.6. Statistics Analysis

Data are presented as mean ± SEM. Statistical analyses were performed using IBM SPSS Statistics 26 software (IBM Corp, Armonk, NY, USA). The experimental data include homogeneity of variance tests and group comparisons using ANOVA. The least significant difference (LSD) method was used for inter-group statistical tests. *p* < 0.05 was considered to be statistically significant.

## 3. Results

### 3.1. NT Supplement Improved Dietary Intake and Body Weight in Aging SAMP8 Mice

To investigate the effect of NTs on weight loss in aging SAMP8 and SAMR1-NC mice, we customized the feed to contain different doses of NTs for long-term feeding. As shown in Table 2, two to four mice in each group died during the intervention, and there was no significant difference in death time and mortality (*p* > 0.05).

As shown in Table 3 and Table 4, the body weight of mice increased at first and then decreased with age. NT supplementation increased the body weight at first because of the higher food intake and decreased the decline in body weight at 12 months old. Removing the NTs from the mice′s food significantly reduced the body weight. Almost all the food intake of mice decreased with age. Table 5 shows the changes in food utilization during the experiment. After 6 months of age, the average food utilization in each group was less than 0, which corresponded to the weight loss of mice after 6 months.

### 3.2. NT Supplement Improved BAT Mass and Decreased the Decline in Total Fat Mass

With an increase in age, both the mass of BAT and total fat mass decreased, and the dietary supplement of 0.3 and 0.6 g/kg NTs maintained the quality to a certain extent (Figure 1A,B,D). It seems that the mass of BAT and total fat mass in the 1.2 g/kg NT supplement group (NTs-H) was slightly lower than that in the SAMP8-NC group, but there was no statistically significant difference. Normalization to total fat mass led to no significant difference between the SAMP8-young-NC, NTs-M, and NMN groups, showing that the dietary intake of NTs helped decrease the decline in total fat tissue.

A previous study showed hypertrophy of adipocytes in SAMP8 mice at the age of 40 weeks [22]. Six-month-old SAMP8 mice showed a lower cold exposure response compared with SAMR1-NC at the same age [23]. However, in our research, there was no significant difference in brown adipocyte size and inflammatory cell infiltration at room temperature (Figure 1C). A large number of vacuolar brown adipocytes and capillaries could be seen under the microscope.

### 3.3. NT Supplement Promoted Antioxidant-Associated Biological Indicators in BAT

We compared the 12 month old mice in the SAMP8-NC group with the 6 month old mice in the SAMP8-young-NC group to explore the dynamic oxidative stress change in BAT with age and the effects of different doses of NTs and NMN (Figure 2). The mice in SAMP8-young-NC presented the highest MDA activity and theirs was significantly higher than those of the 12 month old group (*p* < 0.05). The MDA concentration of the NTs-F and NTs-L groups was significantly lower than that of the SAMP8-NC group. SOD activity decreased with age (*p* < 0.05). The NT intervention groups showed significantly higher SOD activity compared with the SAMP8-NC group (*p* < 0.05). SOD activity in NTs-F was significantly lower than in NTs-L. GSH-Px activity appeared to be independent of age. GSH-Px activity in NTs-F was significantly lower than in SAMP8-NC and SAMR1-NC (*p* < 0.05). GSH-Px activity in NTs-L and NTs-M were significantly higher than in SAMR1-NC (*p* < 0.05). There was no significant difference in CAT activity between the different groups (*p* > 0.05).

### 3.4. NTs Increased Brown Adipocyte Markers in BAT

As shown in Figure 3, the brown adipocyte marker levels in the SAMP8-young-NC group were higher than those in the SAMP8-NC group, showing that the thermogenic ability of BAT decreased remarkably with age (*p* < 0.05). Compared with SAMP8-young-NC, NTs-L appeared to express lower UCP-1 protein levels, but there was no significant difference, which indicates that an extra intake of 0.3 g/kg NTs probably improves the BAT thermogenic capability in aging SAMP8 mice. NTs-M exhibited higher protein levels of transcription factors including PGC-1α and PRDM16. The thermogenic ability of NTs-F dropped overall compared with the other NTs intervention groups. Nicotinamide phosphoribosyltransferase (NAMPT) is a rate-limiting enzyme necessary for NAD+ biosynthesis, which is very important for regulating BAT adaptive thermogenesis, lipolysis, and systemic energy metabolism. Removed NTs from the feed reduced the expression of NAMPT significantly [24]. Peroxisome proliferator-activated receptor-α (PPAR-α) was tightly associated with steatolysis, which declined with age. NTs-L and NTs-M showed higher protein levels of PPAR-α compared with SAMP8-NC.

### 3.5. NTs Enhanced Sirt-1 Protein Level and Might Have Activated AMPK in BAT

To elucidate the mechanism of increased thermogenic ability in NTs-L and NTs-M, we evaluated the protein level of Sirt-1 and the ratio of phosphorylated AMPK (pAMPK) to total AMPK (tAMPK). Consistent with the trends of brown adipocyte markers in different groups, AMPK activity decreased as the mice got older. The AMPK activity was linked to the expression of UCP-1. NTs-L exhibited higher AMPK activity (Figure 4).

NMN is a key NAD+ synthesis intermediate and has been proven to have a protective mitochondrial function by activating Sirt-1, but there was no research showing that extra dietary NTs intake is useful for activating Sirt-1. As our results showed (Figure 4), the protein level of Sirt-1 protein in the NMN group was significantly higher than that in the SAMP8-NC and SAMR1-NC groups. NTs-M showed an excellent capability for activating Sirt-1, which was comparable to that of NMN. Although the NMN group showed higher Sirt-1 activity, we did not observe the promotion of AMPK and thermogenic-related proteins, such as UCP-1, PGC-1αm, and PRDM16, in the NMN group. The results of the experiment indicate that the promotion of thermogenic function in NTs-L and NTs-M was closely related to the activation of the AMPK/Sirt-1 pathway.

## 4. Discussion

BAT acts as anti-diabetes tissue in humans by consuming glucose and free fatty acids [4]. In this study, we found that 0.3 and 0.6 g/kg NT supplementation decreased the decline in BAT mass effectively. Increasing BAT volume by cold stimulation is of positive significance to improve lipid metabolism in adults. Maintaining BAT function in adults is significantly related to systemic lipolysis, free fatty acid circulation and oxidation, and increased insulin sensitivity in adipose tissue [25]. With an increase in age, BAT in rodents showed volume atrophy and mitochondrial function decline. Studies have shown that activated BAT is associated with lower body mass index in preschool children [26], but no sufficient evidence has demonstrated the relationship between BAT and total fat mass in the elderly. Our study found that BAT mass is unconcerned with total fat mass in elderly SAMP8 mice. These results should be taken cautiously.

Aging BAT adipocytes often show an increase in the white adipocyte-like phenotype due to the fact of immune cell infiltration and excessive fatty acid deposition, and these gradually replace BAT adipocytes [6,7] in a process called BAT “whitening”. Six-month-old SAMP8 mice showed lower cold exposure response compared with SAMR1-NC at the same age. In this study, no white-like changes in BAT were observed. Our research was established in an aging mouse model. Considering the viability of elderly mice, we did not give cold stimulation to older mice.

NTs have been proven to be a good antioxidant. Long-term feeding with exogenous NTs inhibited the decrease in age-related antioxidant enzymes and the increase in peroxide in rat serum and significantly increased the average and maximum life span [17]. NTs also improved the oxidative stress in mouse skeletal muscle by increasing the activities of SOD, sorbitol dehydrogenase (SDH), and GSH-Px, reducing the level of MDA and improving the activity of mitochondria, which has an obvious antifatigue effect [27]. Experiments in vitro found that oxidative stress induced by H_2_O_2_ leads to BAT dysfunction by enhancing autophagy, and the use of antioxidants, such as vitamin D, significantly improves the BAT oxidative stress phenotype and induces changes in BAT activity [28]. In this study, we proved that dietary supplementation with NTs improved oxidative stress by reducing MDA production and increasing the activities of antioxidant enzymes SOD and GSH-PX in BAT so as to prevent excessive ROS produced via metabolism from damaging the cell structure and maintaining normal physiological functions.

Although we did not observe morphological changes in brown adipocytes in this study, the expression of UCP-1 in the SAMP8-young-NC group was significantly higher than that in the SAMP8-NC group, which proved that the thermogenic function of BAT gradually declined with age. The extra intake of 0.3 g/kg NTs is helpful to improve the thermogenic function of BAT in mice. UCP-1, which is located in the inner membrane of mitochondria, is a specific protein uniquely expressed in BAT adipocytes and Brite adipocytes. Under the stimulation of cold or norepinephrine released by sympathetic nerves, the increased intracellular cAMP level activates PKA and upregulates the expression of UCP-1. UCP-1 provides a low-resistance pathway for H^+^, and makes H^+^ return directly to the inner mitochondrial membrane, thereby eliminating the mitochondrial proton gradient generated in the process of oxidative phosphorylation and uncoupling with ADP. In this process, brown adipocytes produce heat via non-shivering thermogenesis [5]. ROS is a by-product of mitochondrial respiration. In BAT, the uncoupling of mitochondrial respiration is protective against oxidative stress, because UCPs that uncouple the mitochondrial respiration are involved in controlling uncoupling of ROS production in mitochondria [28]. In addition, UCP-1 is also protective against the production of ROS and resists oxidation through uncoupling. Similar to previous studies, our study found that UCP-1 activity in BAT was negatively correlated with ROS levels in tissues. The higher the expression of UCP-1, the lower the content of MDA [29,30].

Our research observed a decline in brown adipocyte markers in BAT with age. An extra supplement of NTs at 0.6 g/kg powerfully improved the expression of PGC-1α; PGC-1α is a key protein factor regulating the mitochondrial functional network. The protein takes part in the expression of UCP-1 and other thermogenic components, and the activation of thermogenic genes acts as the central transcriptional effector of adrenergic activation of thermogenic adipocytes, directly or indirectly controlling several important metabolic pathways in tissues [31]. PGC-1α acts as a peroxisome proliferator-activated receptor (PPAR)-γ/retinoid X receptor (RXR)α coactivator, being involved in the regulation of mitochondrial processes related to BAT-adaptive thermogenesis and stimulating the production of UCP-1 [32]. PRDM16 is also an important regulator of BAT thermogenic activity. In this research, the expression of PRDM16 also declined with age, and 0.6 g/kg extra NT supplement showed the highest PRDM16 expression in 12 month old mice. PRDM16 is mainly expressed in BAT and plays an important role in the thermogenic differentiation and protein transcription of BAT adipocytes. PRDM16 produces a marked effect by binding to other transcription factors, such as CCAAT/enhancer-binding protein β (c/EBPβ), PPAR-γ, PPAR-α, and PGC-1α, inducing the expression of PGC-1α and UCP-1 to powerfully regulate the differentiation of brown and beige adipocytes [33]. Interscapular BAT is particularly reliant on PRDM16 for maintaining the expression of brown-fat-selective genes during aging [34]. It has been confirmed that knockout of PRDM16 destroys the thermogenic characteristics of BAT adipocytes and increases the expression of white adipose tissue-specific genes [34,35].

Researchers have proven that the activation of AMPK/Sirt-1/PGC-1α helps to promote the biogenesis of mitochondria in adipocytes [36], maintain mitochondrial function, and inhibit oxidative stress, apoptosis, and mitochondrial damage during cerebral ischemia [37] and myocardial ischemia [38]. Enhancing the function of AMPK/Sirt-1 can reduce adipogenesis and inflammation in obese mice induced by a high-fat diet, enhance antioxidant function, and have potential anti-obesity effects [39]. In BAT, the activation of AMPK signaling significantly improves the thermogenic characteristics of brown adipocytes and increases the expression of PGC-1α, PRDM16, and PPAR-α, which are related to thermogenesis and lipolysis abilities, to enhance the energy consumption of adipose tissue [40]. In this study, the activity of AMPK decreased during BAT aging in 6 month old mice compared with 12 month old mice. The AMPK activity of NTs-L was significantly higher than that of SAMR1-NC (*p* < 0.05). It is worth noting that the AMPK activity in NT intervention groups declined gradually with the increase in the dose. AMPK activity in NTs-H was slightly lower than that of the SAMP8-NC group and significantly lower than that of the SAMP8-young-NC group (*p* < 0.05), suggesting that dietary intake of 0.3 and 0.6 g/kg exogenous NTs might be helpful towards activating AMPK, but 1.2 g/kg NTs may be inhibitory. NMN has been proven to be an active synthetic intermediate of NAD+, and it does well in terms of Sirt-1 activation when included in one′s dietary intake [19]. It surprised us that the dietary intake of 0.6 g/kg NTs showed an activation ability of Sirt-1 consistent with NMN, and higher expression of brown adipocyte markers UCP-1, PGC-1α, and PRDM16. Although NMN significantly improves the expression of Sirt-1 in BAT, there was no significant difference between NMN and SAMP8-NC in the expression of PGC-1α and UCP-1.

Different from traditional BAT thermogenic experiments, our long-term feeding research focused on the BAT in aging mice, and we proved that the thermogenic capability of BAT exhibits an age-related decline. Supplementation with 0.3 and 0.6 g/kg NTs might be useful to promote BAT biomarkers by improving the oxidative stress and the alleviation of AMPK/Sirt-1. However, this study did have several limitations. Our study assessed the thermogenic ability by UCP-1 detection. Limited by the low viability of aged mice, we were unable to evaluate the dynamic changes in BAT thermogenic function under cold stimulation. BAT adipocytes in 12 month old SAMP8 mice were shown to be similar to SAMR1-NC at room temperature. Our further experiments will try to prolong the intervention time or increase the cold stimulation in younger mice to comprehensively evaluate the effect of exogenous NTs on BAT.

## 5. Conclusions

Our research shows that the thermogenic ability of BAT was significantly improved by supplementation with 0.3 or 0.6 g/kg of NTs. This effect may be related to the lower oxidative stress and the activation of the AMPK/Sirt-1 pathway.

## Figures and Tables

**Figure 1 nutrients-14-02796-f001:**
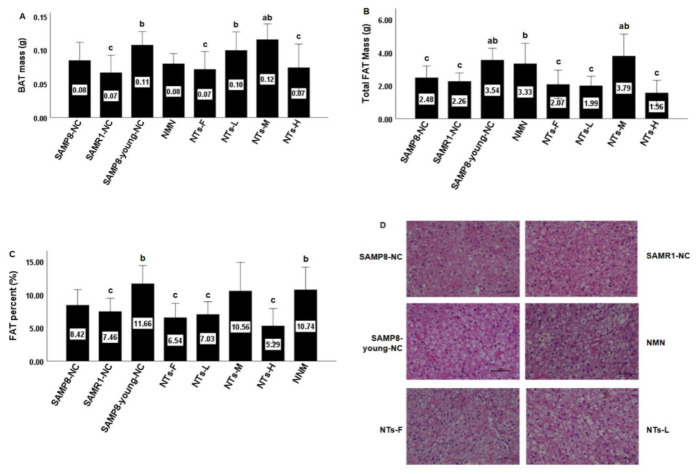
NT supplementation significantly improved BAT mass and total fat mass in NTs-M compared with that in the SAMP8-NC group, showing a similar quality with 6 month old mice: (**A**) interscapular BAT was collected and weighed at the intervention endpoint; (**B**) six mice in each group were randomly chosen for body composition detection at the intervention endpoint and total fat mass is shown as the mean ± SEM; (**C**) fat percent (%) = total fat mass/weight × 100%; (**D**) hematoxylin and eosin (H&E) staining of BAT sections (magnification: 400×). ^a^ Compared with SAMP8-NC, *p* < 0.05; ^b^ compared with SAMR1-NC, *p* < 0.05; ^c^ compared with SAMP8-young-NC, *p* < 0.05. Data were subject to homogeneity of variance tests and group comparisons using ANOVA. The LSD method was used for inter-group statistical tests.

**Figure 2 nutrients-14-02796-f002:**
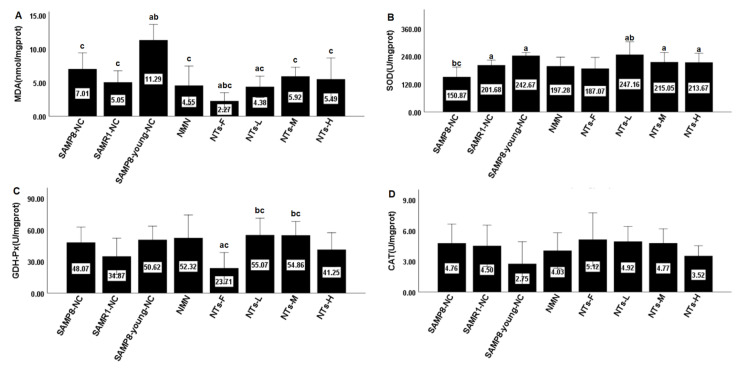
Dietary NT supplement increased the activity of SOD and decreased the concentration of MDA. The BAT in each group was collected and stored at −80 °C. Three mice were used for the Western blot and the remaining 8–10 mice were used for the detection. (**A**) Malondialdehyde was measured via the TBA method. (**B**) Total superoxide dismutase was measured with WST-8. (**C**) Glutathione peroxidase was measured via the colorimetric method. (**D**) Catalase was measured with the visible light method. ^a^ Compared with SAMP8-NC, *p* < 0.05; ^b^ compared with SAMR1-NC, *p* < 0.05; ^c^ compared with SAMP8-young-NC, *p* < 0.05. The data were subjected to a homogeneity of variance test and groups comparisons using ANOVA. The LSDmethod was used for inter-group statistical tests.

**Figure 3 nutrients-14-02796-f003:**
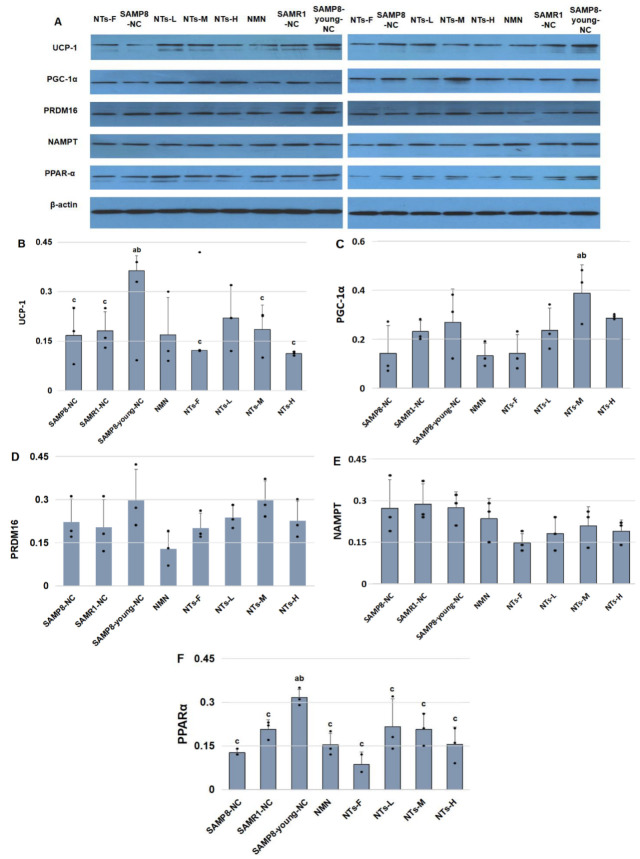
Supplementation with 0.3 and 0.6 g/kg NTs improved the expression of brown adipocyte markers. Three mice were randomly chosen from each group for Western blot detection. β-Actin was used as the loading SAMP8-NC for equal amounts of protein. (**A**) Immunoblots of UCP-1, PGC-1α, PRDM16, NAMPT, and PPAR-α. All the original membranes can refer to Appendix A Appendix A. (**B**) UCP-1/β-actin. (**C**) PGC-1α/β-actin. (**D**) PRDM16/β-actin. (**E**) NAMPT/β-actin. (**F**) PPAR-α/β-actin. Data are shown as the mean ± SEM. ^a^ Compared with SAMP8-NC, *p* < 0.05; ^b^ compared with SAMR1-NC, *p* < 0.05; ^c^ compared with SAMP8-young-NC, *p* < 0.05. Data were tested for homogeneity of variance and groups comparisons using ANOVA. The least significant difference (LSD) method was used for the inter-group statistical test.

**Figure 4 nutrients-14-02796-f004:**
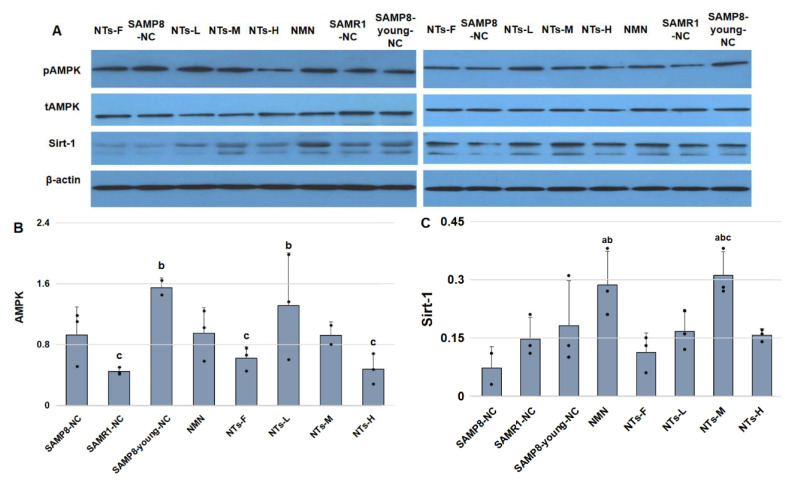
Supplementation with 0.6 g/kg NTs (NTs-M) significantly induced AMPK/Sirt-1 activation. Three mice were randomly chosen from each group for Western blot detection. (**A**) immunoblots of tAMPK, pAMPK, and Sirt-1. All the original membranes can refer to Appendix A Appendix A; (**B**) AMPK activity is shown as pAMPK/tAMPK; (**C**) Sirt-1/β-actin. Data are shown as the mean ± SEM. ^a^ Compared with SAMP8-NC, *p* < 0.05; ^b^ compared with SAMR1-NC, *p* < 0.05; ^c^ compared with the SAMP8-young-NC, *p* < 0.05. Data were tested for homogeneity of variance and groups comparisons using ANOVA. The least significant difference method was used for the inter-group statistical tests.

**Table 1 nutrients-14-02796-t001:** The levels of NTs in the purified and standard food.

Food	Purified Food	Standard Food
C	0	95.4
A	0	348
U	0	714.3
G	0	328.5
Total NT content (mg/kg)	0	1486.2

C, 5′CMP; A, 5′AMP; U, 5′UMP; G, 5′GMP.

**Table 2 nutrients-14-02796-t002:** Animals and treatments.

Groups	Feed	Sample Size	Ages at the Endpoint (m)	Survival Animal Numbers
SAMP8-young-NC	Standard food	12	6	12
SAMP8-NC	Standard food	15	12	12
SAMR1-NC	Standard food	15	12	13
NMN	Standard food + 0.3 g/kg NMN	15	12	12
NTs-F	Purified food, AIN-93M	15	12	11
NTs-L	Standard food + 0.3 g/kg NTs	15	12	13
NTs-M	Standard food + 0.6 g/kg NTs	15	12	13
NTs-H	Standard food + 1.2 g/kg NTs	15	12	12

**Table 3 nutrients-14-02796-t003:** Weight Changes During the Whole Intervention.

Groups	3 Months Old	6 Months Old	9 Months Old	12 Months Old
SAMP8-NC	29.84 ± 1.84	33.27 ± 2.36	31.66 ± 2.59 ^b^	29.78 ± 3.86
SAMR1-NC	30.88 ± 1.53	35.77 ± 2.73	35.78 ± 3.32 ^a^	32.01 ± 3.63
SAMP8-young-NC	30.80 ± 1.95	35.01 ± 2.02		
NMN	30.12 ± 1.77	35.32 ± 3.59	33.45 ± 2.74	31.63 ± 2.80
NTs-F	30.89 ± 3.62	31.89 ± 2.92 ^b^	31.40 ± 2.98 ^b^	27.69 ± 3.09 ^b^
NTs-L	30.32 ± 2.61	33.70 ± 3.93	33.51 ± 4.71	32.16 ± 4.42
NTs-M	30.80 ± 2.59	36.67 ± 4.53 ^a^	35.28 ± 4.48 ^a^	33.28 ± 3.88 ^a^
NTs-H	29.93 ± 1.70	36.46 ± 5.19 ^a^	36.23 ± 4.84 ^a^	32.50 ± 3.12

Changes in body weight were measured every week. ^a^ Compared with SAMP8-NC, *p* < 0.05; ^b^ compared with SAMR1-NC, *p* < 0.05. Data were subject to homogeneity of variance tests and group comparisons using ANOVA. The LSD method was used for inter-group statistical tests.

**Table 4 nutrients-14-02796-t004:** Food intake changes during the whole intervention.

Groups	3 Months Old	6 Months Old	9 Months Old	12 Months Old
SAMP8-NC	35.34 ± 1.54	33.14 ± 4.30	27.86 ± 4.04 ^b^	25.04 ± 2.69
SAMR1-NC	35.63 ± 5.73	33.04 ± 2.18	32.83 ± 2.91 ^a^	26.87 ± 3.42
SAMP8-young-NC	36.69 ± 1.65	33.74 ± 4.18		
NMN	37.11 ± 6.64	33.39 ± 4.14	29.45 ± 3.60 ^b^	28.61 ± 6.15
NTs-F	34.04 ± 8.81	32.17 ± 7.08	29.99 ± 3.77	29.63 ± 3.97 ^a^
NTs-L	36.68 ± 2.49	37.63 ± 4.12 ^ab^	32.89 ± 4.57 ^a^	28.41 ± 2.43
NTs-M	37.75 ± 2.89	37.04 ± 4.98 ^ab^	32.28 ± 4.04 ^a^	25.77 ± 5.64
NTs-H	37.74 ± 2.08	37.49 ± 4.60 ^ab^	30.26 ± 2.89	28.93 ± 4.94

Changes in food intake were measured every week. ^a^ Compared with SAMP8-NC, *p* < 0.05; ^b^ compared with SAMR1-NC, *p* < 0.05. Data were subject to homogeneity of variance tests and group comparisons using ANOVA. TheLSD method was used for inter-group statistical tests.

**Table 5 nutrients-14-02796-t005:** Food utilization changes during the whole intervention.

Groups	3–6 Months	6–9 Months	9–12 Months
SAMP8-NC	0.52 ± 0.25	−0.38 ± 0.35 ^b^	−0.36 ± 0.47
SAMR1-NC	0.71 ± 0.80	−0.46 ± 0.46 ^a^	−0.31 ± 0.52
SAMP8-young-NC	0.39 ± 0.47		
NMN	0.67 ± 0.50	−0.73 ± 0.42 ^a^	−0.18 ± 0.58
NTs-F	0.36 ± 0.86	0.00 ± 0.27	−0.04 ± 0.47
NTs-L	0.62 ± 0.31	−0.50 ± 0.35 ^a^	−0.11 ± 0.31
NTs-M	0.49 ± 0.27	−0.12 ± 0.49	−0.27 ± 0.44
NTs-H	0.53 ± 0.73	−0.62 ± 0.37 ^a^	−0.08 ± 0.31

Three-month food utilization was calculated. Food utilization = weight variation/food intake ×100. ^a^ Compared with SAMP8-NC, *p* < 0.05; ^b^ compared with SAMR1-NC, *p* < 0.05. Data were subject to homogeneity of variance tests and group comparisons using ANOVA. The LSD method was used for inter-group statistical tests.

## Data Availability

Not applicable.

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
