# Peer review of "Exogenous Nucleotides Improved the Oxidative Stress and Sirt-1 Protein Level of Brown Adipose Tissue on Senescence-Accelerated Mouse Prone-8 (SAMP8) Mice"

_nutrients, 2022, doi:10.3390/nu14142796_

Round 1

Reviewer 1 Report

Line# 12

In abstract SAMP8 jumps out of nowhere.

Please describe whats SAMP8 mice.

Abstract needs to be re-written. There are lot of acronyms used without describing what they are.

In fact as I m reading the text, there is no explanation of SAMP8 and SAMR1 anywhere in the paper.

The introduction is merely an assembly of fragmented information which does not follow any flow.

All those treatment acronyms need to be explained. Methods are not described properly.

Reviewer 2 Report

Wang et al investigate the role of exogenous nucleotides on the thermogenesis of BAT in SAMP8. NTs are conditionally essential nutrients and are of various biological functions. This study investigated the effects of NTs on aging in SAMP8 mice. Found that NTs in aged mice improved body weight and BAT mass via AMPK-Sirt-1 pathway.

There is no explanation/definition/rationale of SAMP8 mice, or to the caveats of using senescence accelerated mice. NTs were also not defined – it is not clear if the , NTs-F, NTs-L, NTs-M, NTs-H and NMN refers to the doses of the intervention groups which are listed as NTs-free, 0.3g/kg, 0.6g/kg and 1.2g/kg.

The body weight data is of interest, but pair-feeding should be performed to indicate if the NTs actually affect body weight or if it is in response to altered food intake. Was energy expenditure measured to account for differences?

In addition, it seems as though the authors assume that the decrease in body mass at 12 mos in the NTs-F group is regarded as an ‘improvement’ – but why is it being assumed as a good thing? These mice are not obese – is it just fat mass or a reduction in lean mass as well?

The change in stress-associated indicators in BAT are interesting, but not properly interpreted with regard to changes in body mass. UCP1 westerns are hard to interpret since some of the bands are not clear. This still does not address a change in energy expenditure / food intake / body mass – which could tie the phenotype together very clearly. Are mice more/less cold tolerant? That would help interpret the BAT activity data.

The change in Sirt-1 is relatively minor (0.1 to 0.2) with no units presented – this makes the data difficult to interpret and understand if it is indicative of any physiological change.

The aged brown adipose tissue does not show a ‘whitening’ of the BAT, making the overall conclusions difficult to interpret. Figure 2 does not show changes in histology – can these cells be quantified to support the conclusions?

Minor:

It is confusing that the western data presented is in a different order on the blots than in the bar graphs.

Reviewer 3 Report

 In this study, the authors investigated the effect of dietary nucleotide supplementation on age-induced changes in brown adipose tissue in mice. They found that this supplementation increased  body weight, BAT mass, thermogenic protein expression and improved oxidative stress parameters. The authors conclude that nucleotides can improve age-reduced thermogenesis in BAT by activating the AMPK/Sirt-1 pathway-

Before accepting these conclusions, a number of findings should be clarified

Referee has the following comments on the manuscript:  

Many of the facts necessary to assess the manuscript are not mentioned:

  1. The authors report that the experiments were performed on SAMP8 and SAMR1 mice. These species are not usually used and therefore their characteristics should be stated - especially their sensitivity to obesity, hyperlipidemia, etc.
  2. Why were mice of a different species used as a control?
  3. The numbers of mice in each group are not clearly stated.
  4. Why was the amount of body fat determined by magnetic resonance imaging when accurate measurements of fat depots are made in experimental studies?
  5. Lines 108 – 114: These abbreviations are given here for the first time and should be explained.
  6. Figure 1: lines 136-137? What does n=12 mean? Is this the number of animals in each group (that would mean a total of 168 animals); or is it the number of measurements? How many animals were in each group?
  7. Figure 2: It is not described on the "Y" axis whether the stated weight of BAT and body fat is absolute weight or is based on body weight. What does BAT/g or FAT/g mean?
  8. It is unbelievable how a very small amount of interscapular BAT was sufficient to measure many parameters. For how many animals were the individual parameters examined?
  9. The discussion unnecessarily repeats detailed results, while a summary of the main findings and their novelty is not mentioned.
  10. In addition, many author's statement in the discussion cannot be accepted

- Line 218: „… BAT consumes excess nutrients and is treated as a new target for the treatment of obesity….“ ( Excess nutrients are stored in white adipose tissue. In  BAT it is utilized for thermogenesis.)

- Line 237: „Vitro experiments  found that…“ (It should be in vitro)

- Line 241: „BAT is a natural antioxidant organ….“ (The opposite is true. Higher respiratory chain activity in BAT generates large amounts of free radicals.)

Round 2

Reviewer 2 Report

My comments have been sufficiently addressed.

Author Response

Thank you for your thoughtful comments. Your suggestions is very helpful to us.

Reviewer 3 Report

The authors respected my comments and modified the texts in the manuscript as recommended. Correction of the English language also contributed to the improvement of manuscript quality.

I have only minor comments:

Line 27: I don't understand what it means: “…reduced the comment of malondialdehyde…“.Probably to be “...the concentration…”

Line 136: as far as FAT mass is concerned the authors state FAT / g which is used incorrectly. According to the comments, the authors corrected it in Fig. 2B. And similarly, it must be addressed in this line. i.e. FAT (g).

Line 191: "... showing he dietary intake ..." should be “…showing the dietary

intake..”.

I may be wrong, but I did not find in the manuscript the citation under number 23 in the list of references.

Author Response

Thank you for your careful check. We have checked the text and corrected the mistakes in Line 136, 127 and 191 accordingly.

The references are collated, and the missing citation number 23 was added in Line194-195, showed as "6-month-old SAMP8 mice showed lower cold exposure response compared with SAMR1 at the same age[23]." 

Your thoughtful comments are very helpful to us.